# Subcutaneous Administration of Apolipoprotein J-Derived Mimetic Peptide d-[113–122]apoJ Improves LDL and HDL Function and Prevents Atherosclerosis in LDLR-KO Mice

**DOI:** 10.3390/biom10060829

**Published:** 2020-05-29

**Authors:** Andrea Rivas-Urbina, Anna Rull, Joile Aldana-Ramos, David Santos, Nuria Puig, Nuria Farre-Cabrerizo, Sonia Benitez, Antonio Perez, David de Gonzalo-Calvo, Joan Carles Escola-Gil, Josep Julve, Jordi Ordoñez-Llanos, Jose Luis Sanchez-Quesada

**Affiliations:** 1Cardiovascular Biochemistry Group, Research Institute of the Hospital de Sant Pau (IIB Sant Pau), 08041 Barcelona, Spain; arivas@santpau.cat (A.R.-U.); anna.rull@iispv.cat (A.R.); joile.aldana@gmail.com (J.A.-R.); npuigg@santpau.cat (N.P.); sbenitez@santpau.cat (S.B.); jordonez@santpau.cat (J.O.-L.); 2Biochemistry and Molecular Biology Department, Universitat Autònoma de Barcelona, 08193 Cerdanyola, Spain; 3Hospital Universitari Joan XXIII, IISPV, Universitat Rovira i Virgili, 43005 Tarragona, Spain; 4Molecular Basis of Cardiovascular Risk, Research Institute of the Hospital de Sant Pau (IIB Sant Pau), 08041 Barcelona, Spain; dsantos@santpau.cat (D.S.); farre.nuria88@gmail.com (N.F.-C.); jescola@santpau.cat (J.C.E.-G.); jjulve@santpau.cat (J.J.); 5CIBER of Diabetes and Metabolic Diseases (CIBERDEM), Institute of Health Carlos III, 28029 Madrid, Spain; aperez@santpau.cat; 6Endocrinology Department, Hospital de la Santa Creu i Sant Pau, 08041 Barcelona, Spain; 7Translational Research in Respiratory Medicine, University Hospital Arnau de Vilanova and Santa Maria, IRBLleida, 25198 Lleida, Spain; David.degonzalo@gmail.com; 8CIBER of Respiratory Diseases (CIBERES), Institute of Health Carlos III, 28029 Madrid, Spain

**Keywords:** mimetic peptide, atherosclerosis, lipoprotein function, apolipoprotein J, LDL, HDL, mice

## Abstract

Mimetic peptides are potential therapeutic agents for atherosclerosis. d-[113–122]apolipoprotein (apo) J (d-[113–122]apoJ) is a 10-residue peptide that is predicted to form a class G* amphipathic helix 6 from apoJ; it shows anti-inflammatory and anti-atherogenic properties. In the present study, we analyzed the effect of d-[113–122]apoJ in low-density lipoprotein receptor knockout mice(LDLR-KO) on the development of atherosclerosis and lipoprotein function. Fifteen-week-old female LDLR-KO mice fed an atherogenic Western-type diet were treated for eight weeks with d-[113–122]apoJ peptide, a scrambled peptide, or vehicle. Peptides were administered subcutaneously three days per week (200 µg in 100 µL of saline). After euthanasia, blood and hearts were collected and the aortic arch was analyzed for the presence of atherosclerotic lesions. Lipoproteins were isolated and their composition and functionality were studied. The extent of atherosclerotic lesions was 43% lower with d-[113–122]apoJ treatment than with the vehicle or scramble. The lipid profile was similar between groups, but the high-density lipoprotein (HDL) of d-[113–122]apoJ-treated mice had a higher antioxidant capacity and increased ability to promote cholesterol efflux than the control group. In addition, low-density lipoprotein (LDL) from d-[113–122]apoJ-treated mice was more resistant to induced aggregation and presented lower electronegativity than in mice treated with d-[113–122]apoJ. Our results demonstrate that the d-[113–122]apoJ peptide prevents the extent of atherosclerotic lesions, which could be partially explained by the improvement of lipoprotein functionality.

## 1. Introduction

Cardiovascular disease (CVD) is the leading cause of mortality worldwide; coronary artery disease of atherosclerotic origin is the most common expression of CVD. Plasma lipid levels are a major cause for the development of atherosclerotic lesions and, accordingly, lipid-lowering drugs are the main therapeutic strategy for decreasing cardiovascular risk [1]. However, even with normal lipid levels and in the absence of other classical CVD risk factors, such as hypertension, diabetes, obesity or smoking habits, cardiovascular events can occur. This is known as residual cardiovascular risk, and part of this risk is of lipid origin [2]. This is because the development of atherosclerosis does not come only from high lipid levels but it is also determined by the quality and functionality of lipoproteins. Low-density lipoproteins (LDL) are the main source of the cholesterol that accumulates in the arterial wall. The aggregation of these lipoprotein particles in the subendothelial space is the necessary first step to promote their entrapment in the artery wall and their chemical modification by different mechanisms, including oxidation, lipolysis, and proteolysis [3]. Oxidative modification of lipids in lipoproteins is a key event that triggers the inflammatory process that perpetuates atherosclerosis [4]. Regarding high-density lipoproteins (HDL), these particles play a protective role against atherosclerotic development through several mechanisms that include, among others, the reverse transport of the cholesterol accumulated in arterial lesions and the inhibition of the oxidative modification of LDL [5]. Indeed, increased susceptibility of LDL to oxidation or to aggregation [6], as well as a loss of the protective properties of HDL, has been related to accelerated atherosclerosis [7].

Accordingly, therapies targeting the function and qualitative properties of lipoproteins could help to decrease the residual cardiovascular risk, which persists even after statin treatment [8]. Interest in using peptides has been aroused for a few years. These short peptides are directly derived from the sequence of apolipoproteins (apo) or have another amino acid sequence that mimics their conformation of amphipathic helices [9]. Most studies have analyzed the effects of peptides derived from apolipoprotein A-I (apoA-I) and apolipoprotein E (apoE) [10,11], although other apolipoprotein mimetic peptides, such as apolipoprotein C-II (apoC-II) [12] or apolipoprotein J (apoJ) [13], have also been used. The main mechanisms of action of apoA-I and apoE mimetics are theoretically different, since apoE mimetics mediate hepatic clearance of atherogenic apoB-containing lipoproteins from the circulation via interaction with lipoprotein receptors, whereas apoA-I mimetics mainly stimulate the reverse cholesterol transport pathway [11]. Additionally, apoE mimetics enhance cholesterol efflux from cholesterol-loaded macrophages [14]. However, both groups of mimetics have also shown anti-inflammatory and antioxidant properties, avoiding the oxidation of LDL by sequestering lipoperoxides [11]. In both cases, the administration of these peptides has been successful for atherosclerosis prevention in animal models [15,16,17]. However, clinical trials conducted in humans with HDL mimetics have failed to demonstrate a clinical benefit [18]. One possibility that could explain the lack of positive results is the fact that targeting only HDL function could not be enough to prevent the development of atherosclerosis in humans, since, in contrast to mice, LDL is the main transporter of cholesterol in humans. So far there are no data from clinical trials carried out with apoE mimetics.

Despite the relevance of LDL aggregation in the earliest stages of atherogenesis, no specific therapeutic strategies targeting this process have been successfully developed. Compelling evidence supports the potential therapeutic use of mimetic peptides as potential agents to prevent LDL aggregation. Indeed, 4F, a mimetic of apoA-I, was reported to inhibit LDL aggregation in vitro [19]. Another peptide that has been shown to inhibit LDL aggregation in vitro is d-[113–122]apoJ, a mimetic derived from the sequence of apoJ [20]. This peptide is a 10-residue peptide spanning the predicted class G* amphipathic helix 6 from apoJ and has been successfully used to retard atherosclerosis in apoE-knockout mice, which was attributed to its capacity to improve the anti-inflammatory properties of HDL [13]. Our group previously described that both the whole molecule of apoJ [21] and the peptide d-[113–122]apoJ [22] inhibit SMase-induced and spontaneous LDL aggregation in vitro. Based on these observations, the aim of the present study was to analyze the effect of the d-[113–122]apoJ peptide on lipoprotein function and the development of atherosclerosis. For this purpose, we used LDLR-KO mice fed an atherogenic diet, a model that, in contrast to other murine models of accelerated atherosclerosis, generates abundant lipoprotein particles similar in size and density to human LDL [23]. This allowed us to test not only the effect of treatment on HDL function but also on LDL susceptibility to aggregation and their relationship with atherosclerosis.

## 2. Materials and Methods

### 2.1. Animal Study Design

LDLR-KO mice from the C57BL/6 background were purchased from Jackson Laboratories (#002207). Mice were housed in a controlled temperature environment (22 °C), exposed to a 12-h light/dark cycle, and food and water were provided ad libitum. For this study, we used 15-week-old female mice fed a Western diet (WD) (TD.88137, Harlan Teklad, Madison, WI, USA, containing 21% fat and 0.2% cholesterol) for eight weeks. The reason for using females is that the development of atherosclerosis is higher in female than in male mice [24]. Animals were randomly assigned to one of three groups and treated for eight weeks with the d-[113–122]apoJ peptide (*n* = 12), scrambled peptide (*n* = 8) or vehicle (control, *n* = 8). Peptides (10 µg/g body weight) were administered subcutaneously three days per week (200 µg in 100 µL of saline). Weight control and food intake were monitored for all experimental groups (Appendix A). Animals were sacrificed after eight weeks of treatment. After euthanasia, fasting blood, heart, and liver were collected. Appendix A depicts a flow chart of the experimental design. All animal procedures were conducted in accordance with published regulations and were approved by the Institutional Animal Care Committee of the Institut de Recerca of the Hospital de la Santa Creu i Sant Pau (ref. 9375).

### 2.2. Peptides

The amino acid sequence of the d-[113–122]apoJ peptide is Ac-LVGRQLEEFL-NH2. This peptide forms a class G* amphipathic helix in the presence of lipids [20]. The inactive control scrambled peptide (Sc-d-[113–122]apoJ) has the same overall amino acid composition as d-[113–122]apoJ but in a sequence that prevents G* amphipathic helix formation (Ac- LRGVQLLEFE-NH2). Both peptides were obtained from Caslo (Kongens Lyngby, Denmark). The use of D-peptides guarantees the resistance to proteolysis in the blood since these are less susceptible to proteolytic degradation in the digestive tract or inside cells [25]. Navab et al. reported that d-[113–122]apoJ was poorly associated with lipoproteins and mostly remained in the fraction of lipoprotein-deficient plasma during the first hours in blood. Then, the peptide progressively tended to associate with lipoprotein fractions, showing a prolonged residence-time since D-[113–122]apoJ was cleared from plasma much more slowly than peptides derived from apoA-I [13].

### 2.3. Lipid Analysis

Plasma lipid profile and lipoprotein composition of major lipids were measured by commercial methods adapted to a Cobas 6000/c501 autoanalyzer (Roche Diagnostics, Basel, Switzerland). Total cholesterol, triglycerides, aspartate transaminase (AST), and alanine transaminase (ALT) reagents were obtained from Roche Diagnostics. Total phospholipids, non-esterified fatty acids (NEFAs), and free cholesterol reagents were obtained from Wako Chemicals (Richmond, VA, USA). Cholesterol, triglycerides, and phospholipid content in livers was determined by commercial methods adapted to a Cobas 6000/c501 autoanalyzer (Roche Diagnostics, Basel, Switzerland) after solvent extraction [26]. Briefly, liver lipids were extracted with isopropyl alcohol-hexane (2:3, *v*/*v*). The lipid layer was evaporated and resuspended in 0.5% (*w*/*v*) sodium cholate (Serva, Heidelberg, Germany).

### 2.4. Lipoprotein Isolation and Characterization

VLDL (density <1.019 g/mL), LDL (density 1.019–1.063 g/mL), and HDL (density 1.063–1.210 g/mL) were isolated from mice plasma by sequential ultracentrifugation, using potassium bromide for density adjustment, at 100,000× *g* for 24 h with an analytical fixed-angle rotor (50.3, Beckman Coulter, Fullerton, CA, USA) [27]. Lipoprotein composition of major lipids was measured by commercial methods adapted to a Cobas 6000/c501 autoanalyzer, as described above. The protein content of each lipoprotein was assessed by the PierceTM BCA Protein Assay (Thermo Fisher Scientific, Rockford, IL, USA).

### 2.5. Susceptibility to Oxidation of Lipoproteins

Mice LDL and HDL susceptibility to copper-induced lipid oxidation and mice HDL capacity to inhibit the oxidative modification of human LDL were measured by monitoring the formation of conjugated dienes at 234 nm, for 7 h at 37 °C in a BioTek Synergy HT spectrophotometer (BioTek Synergy, Winooski, VT, USA) [28]. Briefly, LDL was dialyzed in phosphate-buffered saline by gel filtration chromatography on PD-10 columns (GE Healthcare, Chicago, IL, USA) and HDL was dialyzed by Dialysis Cassettes G2 (Thermo Fisher Scientific, Rockford, IL, USA). Oxidation was started by adding 2.5 μmol/L CuSO_4_ in wells containing mice LDL, mice HDL alone or mice HDL in the presence of human LDL (0.1 mmol/L cholesterol in all cases). The maximum slope during the propagation phase of the kinetics was the parameter used to estimate the susceptibility of lipoproteins to oxidation. The antioxidant capacity of HDL was expressed as the ability to decrease the slope of the oxidation kinetics of human LDL alone after the subtraction of the kinetics of mice HDL alone, as previously described [29].

### 2.6. Susceptibility to LDL Aggregation

Aggregation of mice LDL was induced by SMase lipolysis. Mice LDL (0.6 mmol/L cholesterol) dialyzed in phosphate-buffered saline was incubated at 37 °C for 2 h with SMase from *Bacillus cereus* sp. (Sigma Diagnostics, Livonia, MI, USA) with a final concentration of 50 mU/mL in the presence of 2 mM CaCl_2_ and 2 mM MgCl_2_. Size-exclusion chromatography (SEC) was performed for monitoring aggregation using a Superose 6 Increase 5/150 GL column in an AKTA-FPLC system (GE Healthcare), as described [21]. Two hundred μL of LDL was injected in the column, eluted at a flow rate of 0.3 mL/min, and peaks were detected at 280 nm. This chromatography discriminates two peaks of LDL, eluting first aggregated LDL (2.5 mL) and later monomeric LDL (3.1 mL).

### 2.7. Electronegativity of LDL

The electric charge of LDL was determined by anion exchange chromatography. Chromatography was conducted in an ÄKTA–FPLC system (GE Healthcare), using a MonoQ 5/50 GL column (GE Healthcare). LDL isolated from mice (0.4 mmol/L cholesterol) was dialyzed against buffer A (Tris 10 mmol/L, EDTA 1 mmol/L, pH 7.4) and 100 μL was injected in the column at a flow rate of 2 mL/min. Two fractions of LDL (native or LDL(+) and modified or LDL(-)) were separated with a stepwise NaCl gradient using buffer A and buffer B (same composition as buffer A containing 1 M NaCl). This gradient was similar to that previously described [30], except that both LDL fractions in mice eluted at a higher ionic strength than human LDL. Accordingly, LDL(+) eluted at 35% buffer B while LDL(-) eluted at 70% buffer B. The percentage of these fractions was calculated from the 280 nm peak area integration.

### 2.8. Cholesterol Efflux Capacity of HDL

In vitro cellular cholesterol efflux induced by HDL was determined using [3H]cholesterol-labeled J774A.1 mouse macrophages (ATCC^®^ TIB67™, Manassas, VA, USA), as previously described [31]. Briefly, 1.5 × 10^5^ cells/well were seeded in 6-well plates and allowed to grow for three days in RPMI 1640 medium containing 2 mM L-Glutamine (Pan Biotech, Aidenbach, Germany) complemented with 10% fetal bovine serum (FBS) (Pan Biotech) and 100 U/mL penicillin/streptomycin (Dominique Dutscher, Brumath, France). After that, cells were labeled for 60 h in the presence of one µCi/well of [1α,2α(n)-3H]cholesterol (GE Healthcare, Little Chalfont, UK) and 5% FBS. The cells were then equilibrated with 0.2% bovine serum albumin (BSA) in medium overnight and incubated for 4 h with mice HDL (25 µg/mL protein), previously isolated by ultracentrifugation and dialyzed in phosphate-buffered saline, as described above. Radioactivity was measured in both the medium and the cells, and the percentage of cholesterol efflux was calculated.

### 2.9. Evaluation of Atherosclerotic Lesions

The proximal aorta in each mouse was isolated, embedded in optimal cutting temperature (OCT) (VWR, Fontenay-sous-Bois, France), and immediately flash frozen. The proximal aorta was serial sectioned, and eight sections were stained for lipids with Oil red O. Atherosclerotic lesion severity was expressed as the area of positive Oil red O staining in four sections separated by 80 µm. The first section was taken 80 µm distal to the point placed between the end of the aortic sinus and the beginning of the aorta. The lesion area (surface area stained with Oil red O) was quantified using AxioVision V 4.8.1.0 image analysis software (Zeiss, Oberkochen, Germany).

### 2.10. Quantitative RT-PCR Analyses

Total liver RNA was obtained using TRIzol LS reagent (Invitrogen, Carlsbad, CA, USA) and was purified with an EZ-10 DNAaway RNA Miniprep Kit (Bio Basic, Markham, ON, Canada). cDNA was generated using EasyScript First-Strand cDNA Synthesis SuperMix (Transgen Biotech, Beijing, China) and quantitative real-time PCR amplification was performed using the GoTaq(R) Probe qPCR Master Mix (Promega, Madison, WI, USA). Specific TaqMan probes (Applied Biosystems, Foster City, CA, USA) were used for *Tnfα* (Mm99999068_m1), *Ccl2* (Mcp1, Mm0441242_m1), *Cd36* (Mm00432403_m1), and *Cd68* (Mm03047343_m1); *Gapdh* (Mm99999915_g1) was used as the control housekeeping gene. Reactions were run on a CFX96TM Real-Time System (Bio-Rad, Hercules, CA, USA) according to the manufacturer’s instructions. The relative mRNA expression levels were calculated using the ΔΔ*C*_t_ formula.

### 2.11. Statistical Methods

GraphPad Prism 6.0 software (GraphPad, San Diego, CA, USA) and the statistical software package R version 3.5.2 (www.r-project.org) were used to perform statistical analyses. Student’s *t*-test was used to compare the differences between two groups. Two-way ANOVA with Tukey’s multiple comparisons post-test was used to compare more than two groups. To analyze the association between atherosclerosis and parameters of lipoprotein function, Spearman’s rho correlation was used, considering all variables as non-parametric. Linear regression analyses were performed to explore potential confounding. Data are expressed as the standardized beta coefficient (β). A *p*-value < 0.05 was considered statistically significant.

## 3. Results

### 3.1. Subcutaneous Administration of d-[113–122]apoJ Reduces Atherosclerosis

After the atherogenic diet, d-[113–122]apoJ-treated mice developed less extensive aortic atherosclerosis than control and scramble-treated mice. Figure 1a shows the mean size of atherosclerotic lesions in four consecutive sections of the proximal aorta. The area of proximal atherosclerosis in d-[113–122]apoJ-treated mice was 40%–45% lower (55,305 ± 19,155 µm^2^/section) than in control (92,549 ± 18,880 µm^2^/section) or scramble (97,144 ± 36,140 µm^2^/section) mice. Representative examples of the lesions observed in each group are shown in Figure 1b (Control), Figure 1c (Scramble), and Figure 1d (d-[113–122]apoJ peptide). These images also showed that mice treated with the apoJ peptide developed less advanced atherosclerotic lesions and more restricted aortic valve attachments compared with the larger lesions that extended to the free aortic wall in both Control and Scramble mice.

### 3.2. d-[113–122]apoJ Did Not Alter the Lipid Profile

Table 1 shows the main biochemical parameters of LDLR-KO mice after eight weeks on a Western diet. This atherogenic diet promoted strong hyperlipidemia with very high levels of total plasma cholesterol, triglycerides, and phospholipids in LDLR-KO mice. Most of the cholesterol was associated with VLDL and LDL, and approximately only 7–8% of cholesterol was transported by HDL when lipoproteins were isolated by ultracentrifugation. However, no statistical difference in the parameters of the lipid profile was observed between groups, with very similar levels of triglycerides, phospholipids, and cholesterol (Table 1). This was confirmed when the lipoprotein profile was determined by size exclusion chromatography (Figure 2). The chromatographic profiles overlapped in the three groups based on the absorbance at 280 nm or the cholesterol in the collected fractions.

Regarding liver parameters, no evidence of hepatic injury was observed in any of the groups, and the liver weights were similar among groups (Table 1). Accordingly, the content of cholesterol or triglycerides in livers was also similar in all groups (Table 1).

Regarding the composition of isolated lipoproteins, no statistically significant difference was detected between groups with respect to LDL. The VLDL of the control group presented an increased triglyceride content compared with scramble or peptide groups (Figure 3). We also observed an increased content of free cholesterol in HDL of mice treated with the peptide compared to control and scramble groups.

### 3.3. d-[113–122]apoJ Favorably Influences the Oxidative Properties of Lipoproteins

The oxidizability of LDL was determined by monitoring the kinetics of LDL oxidation induced by CuSO_4_, measuring the conjugated diene formation at 234 nm. Data are expressed as the mean increase in absorbance per minute, instead of the classical lag phase time because, unlike human LDL, murine LDL does not follow the characteristic sigmoidal curve but rather presents a continuous increase in absorbance. This assay showed that all groups had similar LDL susceptibility to oxidation (Figure 4a,b). HDL susceptibility to oxidation was measured from the maximum slope of the oxidation kinetics. In contrast to LDL, HDL susceptibility to oxidation was significantly lower in mice treated with d-[113–122]apoJ than in control or scramble mice (Figure 4c,d). Moreover, HDL from d-[113–122]apoJ-mice presented a higher capacity to retard the oxidation of human LDL than HDL from the other two groups (Figure 4e,f), as can be deduced from the decreased increase in absorbance at a maximum slope of the propagation phase. These data indicate that although the oxidative properties of LDL were similar among groups, the antioxidative action of HDL was potentiated in d-[113–122]apoJ-treated mice.

To test if the observed effects of the d-[113–122]apoJ peptide on lipoprotein oxidation were due to a direct action of the peptide on the surface of lipoproteins, we performed oxidation kinetics using human LDL, HDL or LDL + HDL under the same conditions described above, adding increasing amounts of d-[113–122]apoJ (peptide/apoB or peptide/apoA-I molar ratios of 1/10, 1/1, and 10/1). No direct effect of the peptide was observed (Appendix A). This suggests that the ability of d-[113–122]apoJ to increase the antioxidant capacity of HDL could be due to secondary alterations in HDL metabolism rather than to a primary effect of the peptide.

### 3.4. d-[113–122]apoJ Decreases the Susceptibility of LDL to Aggregation

The aggregation of LDL was induced by incubation with SMase and was measured by size-exclusion chromatography, as indicated in the Methods. The basal proportion of aggregated LDL particles (without SMase treatment) was similar in all groups. LDL SMase-induced aggregation assay showed that LDL isolated from mice treated with the d-[113–122]apoJ peptide was less prone to aggregate than LDLs isolated from the control or scramble groups (Figure 5a). Figure 5b shows a representative size-exclusion chromatogram after the induction of LDL aggregation by SMase.

### 3.5. Electronegativity of LDL Decreased upon d-[113–122]apoJ Treatment

The electronegativity of LDL was determined by anion-exchange chromatography separating LDL subfractions according to their electric negative charge, as indicated in the Methods. LDL from d-[113–122]apoJ-treated mice had a lower proportion of the most electronegative subfraction (LDL(-) than control or scramble-treated mice (Figure 6). This observation was not due to a direct effect of d-[113–122]apoJ on the LDL particle since in vitro addition of the peptide to human LDL did not increase the electric charge of LDL (data not shown).

### 3.6. d-[113–122]apoJ Improves the Cholesterol Efflux Capacity of HDL

The ability of HDL to enhance cholesterol efflux from [3H]cholesterol-labeled J774A.1 mouse macrophages was measured in vitro, as described in the Methods. HDL from mice treated with d-[113–122]apoJ exhibited a higher efflux capacity compared to scramble or control groups (Figure 7).

### 3.7. d-[113–122]apoJ Reduces the Hepatic Expression of Inflammation-Related Genes

The administration of d-[113–122]apoJ had no effect on liver size or lipid content (Table 1). However, the expression of the inflammatory mediators *Tnfα* and *Ccl2* was lower than in the other mice groups (Figure 8). Additionally, the expressions of *Cd36* and *Cd68* scavenger receptors were significantly lower in mice treated with the d-[113–122]apoJ peptide than in control or scramble-treated mice. As a whole, these findings suggest an anti-inflammatory effect of the subcutaneous administration of d-[113–122]apoJ at the hepatic level.

### 3.8. Improvements in HDL Oxidation and LDL Electronegativity Are Associated with Atherosclerosis Burden

Spearman’s rho correlation was used to analyze the statistical correlations between the atherosclerosis burden and the different parameters of lipoprotein function. Figure 9 shows correlations of atherosclerosis with the main parameters of lipoprotein function. We found statistically significant correlations between the area of atherosclerotic lesions and two parameters of lipoprotein function, the susceptibility to oxidation of HDL and the electronegativity of LDL. These observations suggest that both parameters could be related to the development of atherosclerosis, and that the improvement promoted by the subcutaneous administration of the d-[113–122]apoJ peptide in these functions could be involved in the reduction of atherosclerotic lesions. Linear regression analysis confirmed that HDL susceptibility to oxidation was independently associated with atherosclerosis after adjusting for confounding (Appendix A). Regarding LDL electronegativity, the association with atherosclerosis was not as strong, with a loss of significance with some variables. Probably, the relatively weak association between LDL electronegativity and atherosclerosis was due to the small sample size (*n* = 4 in each group), but a clear trend of statistical significance was observed.

Besides the correlations observed with atherosclerosis burden (Figure 9), other correlations among lipoprotein function parameters were detected (Appendix A). HDL susceptibility to oxidation was associated with the antioxidant capacity of HDL and also with LDL susceptibility to oxidation. This observation probably reflects the presence of an increased oxidative stress in blood of mice, which was improved by the administration of the peptide. In addition, the electronegative charge of LDL correlated negatively with LDL susceptibility to aggregation.

Alternatively, to analyze if the effect of the treatment was associated with some of the variables, we also conducted a linear regression analysis using the treatment as a covariate and sequentially adding the rest of the variables (Appendix A). A slight attenuation was observed for both HDL susceptibility to oxidation and LDL electronegativity. This analysis concurs with the results shown in Appendix A and strengthens the putative role of the improvement of HDL susceptibility to oxidation and LDL electronegativity, mediating the effect of the peptide.

### 3.9. The Atherosclerosis Burden Correlates with Hepatic Inflammation

Spearman’s rho correlation was also used to analyze the statistical correlations between the atherosclerosis burden and parameters of hepatic inflammation (Figure 10). We found statistically significant correlations between the area of atherosclerotic lesions and the hepatic expression of *Ccl2*, *Tnfα*, and *Cd68*, and a positive trend with *Cd36*. These observations suggest that the development of atherosclerosis is related to the degree of inflammation at a hepatic level.

## 4. Discussion

Our data show that subcutaneous administration of the peptide d-[113–122]apoJ for eight weeks retards the development of atherosclerosis in LDLR-KO mice fed atherogenic diet. Since the 1980s, short synthetic peptides with sequences that mimic those found in natural apolipoproteins have been studied and used to prevent atherosclerosis [9,32]. Several mimetics of apoA-I and apoE have succeeded in preventing atherosclerosis in animal models [10,11,33]; however, human clinical trials involving ApoA-I apolipoprotein mimetic peptides have failed to prove a clinical benefit [18,32]. This could be due to the marked differences in the lipoprotein metabolism between humans and mice. Thus, while LDL plays a preponderant role in cholesterol transport in humans, in mice, most of the cholesterol is transported in blood by HDL. Furthermore, LDL cholesterol plasma levels are not the only risk factor, and besides its plasma concentration, qualitative characteristics of LDL determine its atherogenicity [34,35]. The same occurs with HDL plasma levels, since it has been shown that only increasing its concentration does not decrease cardiovascular risk if its antiatherogenic qualitative properties are not improved [36,37].

We hypothesized that one peptide acting simultaneously on the function of both HDL and LDL could retard the onset or reduce the development of atheromatous lesions. Based on previous studies, we hypothesized that the peptide d-[113–122]apoJ could accomplish protective actions by improving both HDL and LDL function. On one hand, the d-[113–122]apoJ mimetic peptide has in vitro protective effects on human LDL aggregation, a crucial LDL event for the development of atherosclerosis [22]. On the other hand, this peptide was proven to retard atherosclerosis in apoE KO mice by improving the anti-inflammatory properties and the cholesterol efflux capacity of HDL [13]. The novelty of the present study is the analysis of the effect of d-[113–122]apoJ in the LDLR-KO mice which showed higher LDL levels and enhanced atherosclerosis when fed a western-type diet.

In accordance with this hypothesis, data presented in the current study show that the d-[113–122]apoJ peptide induced a reduction of 40% of atherosclerotic lesion areas under severe hyperlipidemic conditions, which is related to a qualitative increase of the antiatherogenic actions of HDL, as well as the improvement of the atherogenic properties of LDL. This action was not caused by quantitative changes in the lipoprotein profile since d-[113–122]apoJ did not exert any effect on the plasma concentration of lipoproteins. Our data show that d-[113–122]apoJ had only minor effects on the main components of lipoproteins; however, a relevant change was an increase in the content of free cholesterol in d-[113–122]apoJ-treated mice. A higher content of free cholesterol on the surface of lipoproteins is known to reduce their oxidizability by increasing the lipid packaging [38,39]; this could therefore contribute to the improved antioxidant properties observed in HDL from d-[113–122]apoJ-treated mice. Another positive effect of d-[113–122]apoJ administration on HDL was a rise in its capacity to promote the efflux of cholesterol from macrophages. Both observations agree with the previous findings by Navab et al., who previously reported that oral administration of d-[113–122]apoJ prevented atherosclerosis development in apoE-KO mice [13]. According to our findings, these authors also reported an improvement of the anti-inflammatory properties and the cholesterol efflux capacity of HDL from treated mice. However, in comparison with the work by Navab et al., our study presents some novelties. First, in our case, LDLR-KO mice were fed an atherogenic diet, whereas in the study of Navab et al., apoE-KO mice were fed a chow diet. Consequently, the levels of plasma cholesterol were much higher in our model and presumably, the development of lesions should be higher. Despite the high levels of cholesterol, the administration of the peptide in our model was able to reduce the extent of atherosclerotic lesions.

Another important difference between our study and the work of Navab and collaborators is the fact that LDLR-KO mice presented high LDL levels whereas in apoE-KO mice, most of apoB-containing particles were in the size and density ranges of VLDL. Thus, our study allowed us to evaluate the effect of d-[113–122]apoJ on the atherogenic properties of murine LDL. Although we did not observe an effect of peptide administration on LDL composition or susceptibility to oxidation, other atherogenic characteristics of LDL were improved. First, we found that d-[113–122]apoJ treatment prevented the aggregation of LDL induced in vitro by SMase, suggesting LDL particles would be less prone to aggregation in vivo, and thereby could have a protective role against the subsequent subendothelial lipoprotein retention. Second, the electronegativity of LDL particles was lower in mice treated with d-[113–122]apoJ than in the control or scramble groups. Although the atherogenic and inflammatory properties of electronegative LDL fractions in mice are not as well established as in humans [40], an increased proportion has been reported in rodents suffering accelerated atherosclerosis [41,42]. The finding of a negative significant correlation between LDL electronegativity and aggregability is paradoxical since studies conducted in humans show that the electronegative fraction of LDL (LDL(-)) is prone to aggregation [40]. However, this aspect has not been previously studied in mice, and the putative association between electronegativity and aggregability of human LDL could be different in murine LDL.

The role of the liver as a crucial driver of inflammation in cardiovascular disease [43] guided us to determine the expression of genes known to be involved in inflammation. Because mRNA levels of *Tnfα*, *Ccl2*, and *Cd68* are commonly elevated in hepatic inflammation, they were selected to assess the potential anti-inflammatory response by ApoJ mimetic peptide in this organ. The hepatic expression of inflammatory mediators was decreased in those mice treated with the d-[113–122]apoJ peptide, as indicated by the reduced expression of *Tnfα* and *Ccl2* (also known as *mcp1*) genes. In line with these observations, the decreased expression of *Cd68* in d-[113–122]apoJ-treated mice indicates lower infiltration of macrophages in the liver. On the other hand, the expression of the scavenger receptor CD36 was significantly lower in those mice treated with d-[113–122]apoJ, which indicates that the increased liver *Cd36* expression associated with enhanced hepatic fatty acid uptake and triglyceride accumulation, as a consequence of a high-fat diet [44], is prevented by the administration of the d-[113–122]apoJ peptide. Overall, these changes in gene expression indicate that the administration of d-[113–122]apoJ improves some abnormalities induced by an atherogenic diet in liver inflammation.

Our results strongly indicate that d-[113–122]apoJ acts through several mechanisms on the function of different lipoproteins and also at the hepatic level, and the combination of these effects would be responsible for atheroprotection. Univariate correlation and linear regression analyses suggest that the main determinants of the decrease of atherosclerosis development by d-[113–122]apoJ administration were the HDL susceptibility to oxidation and LDL electronegativity. However, the specific mechanisms by which the peptide exerts this protective action are not well understood. Regarding LDL, we previously reported that d-[113–122]apoJ protects LDL from in vitro aggregation induced by different mechanisms, including spontaneous and SMase-induced aggregation [22]. We proposed that this effect was mediated by the binding of the peptide to hydrophobic patches in the lipoprotein surface generated during the aggregation process, avoiding the interaction of different LDL particles. Therefore, this direct effect could be the cause of decreased susceptibility to aggregation observed ex vivo. However, the reason for decreased negative electric charge of LDL is unclear. It could be due to the increased antioxidant capacity of HDL preventing the formation of oxidized LDL. Another mechanism that could be related to the reduction of LDL electronegativity is the decrease of liver inflammation, which could reflect a reduction in the systemic inflammation status. It is known that inflammation favors different mechanisms of lipoprotein modification such as oxidative stress, or increased expression at the arterial wall level of lipases and proteases [45]. On the other hand, Navab et al. demonstrated that d-[113–122]apoJ reduces the content of lipoperoxides in lipoproteins from monkeys treated with this peptide, this property being shared by other amphipathic peptides [13]. Thus, this lipoperoxide-sequestering activity of d-[113–122]apoJ could be key to explain the decreased susceptibility to oxidation of HDL, as well as its increased antioxidative properties. At the same time, decreased oxidation of HDL results in maintaining the activities of several enzymes and apolipoproteins involved in the reverse cholesterol transport, such as apoA-I, lecithin:cholesterol acyltransferase, cholesteryl ester transfer protein, paraoxonase or platelet-activating factor acetylhydrolase. Of note, the correlation analysis showed that the parameters mainly related to the reduction in atherosclerosis development were the susceptibility to the oxidation of HDL and the electronegativity of LDL, which gives special relevance to maintain the physiological properties of both lipoproteins. In addition, the development of atherosclerosis was also related to markers of liver inflammation. However, the correlation analysis should be considered with caution since the number of analyzed samples is small to draw more consistent conclusions, and other factors related or not with lipoprotein function or hepatic inflammation could be involved in the anti-atherogenic action of d-[113–122]apoJ.

## 5. Conclusions

In summary, our data show that d-[113–122]apoJ administration decreases the extent of atherosclerotic lesions in close association with simultaneous improvements in the functionality of both HDL and LDL, and with lower inflammation at the hepatic level. These effects were observed under strong hypercholesterolemic conditions, suggesting that d-[113–122]apoJ administration could be a useful therapy against diet-induced atherosclerosis.

## Figures and Tables

**Figure 1 biomolecules-10-00829-f001:**
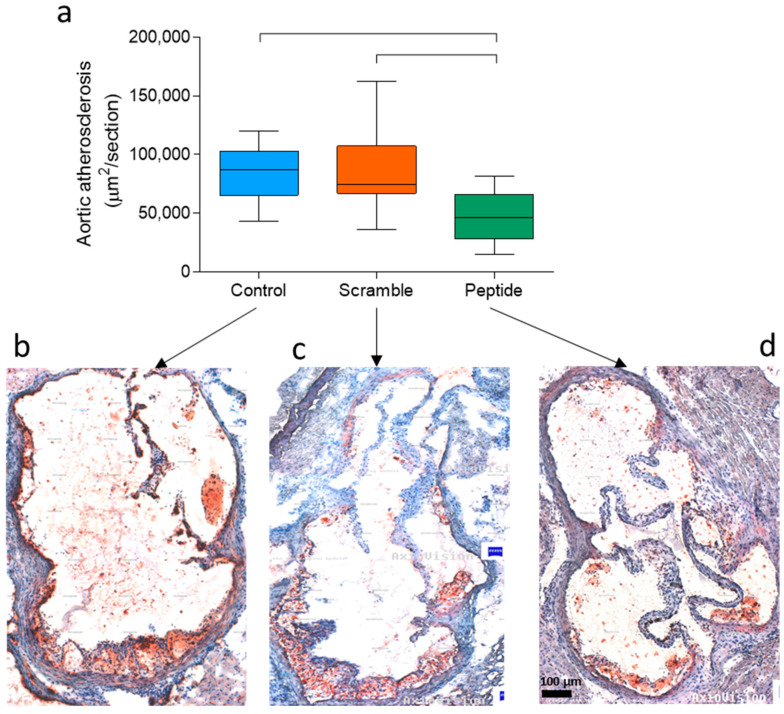
Evaluation of atherosclerotic lesions. Fifteen-week-old female low-density lipoprotein receptor knockout mice (LDLR-KO) mice fed an atherogenic Western-type diet were treated for eight weeks with the d-[113–122]apoJ peptide (*n* = 12), scrambled peptide (*n* = 8) or vehicle (control, *n* = 8). Peptides were administered subcutaneously three days every week (200 µg in 100 µL of saline), as described in the Methods. After euthanasia, hearts were collected, embedded in optimal cutting temperature (OCT) compound, and the aortic arch was sectioned every 20 µm. Lesion areas were stained with Oil red O, and the surface size was quantified using AxioVision image analysis software. (**a**) Lesion size in four consecutive sections of the proximal aorta. Data are shown as box-plot graphs. Bars indicate *p* < 0.05 vs. control or scramble mice. (**b**–**d**) are representative lesions of mice treated with vehicle, scramble or d-[113–122]apoJ peptide, respectively.

**Figure 2 biomolecules-10-00829-f002:**
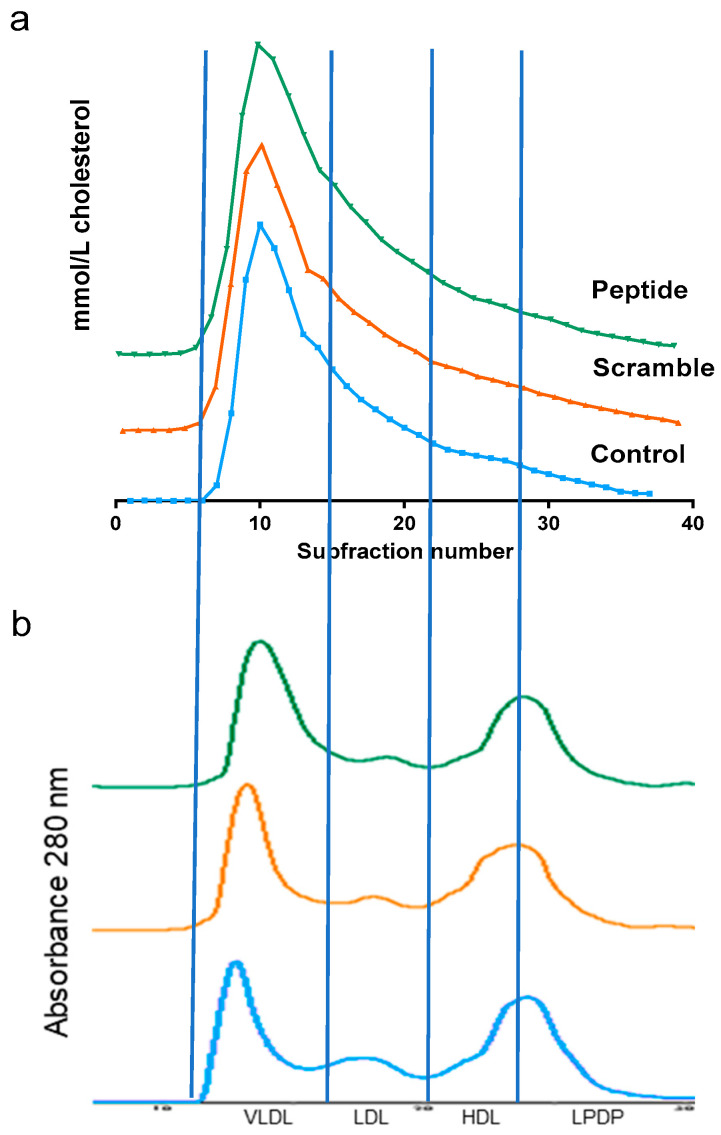
Representative lipoprotein profile determined by size exclusion chromatography, as described in the Methods. Briefly, 400 µL of plasma was injected in a Superose 6 column and eluted with an isocratic buffer at 0.5 mL/min. Fractions (100 µL) were collected and tested for the cholesterol content. (**a**) Cholesterol profile. (**b**) Ultraviolet (UV) 280 nm profile. Vertical lines indicate the volumes at which VLDL, LDL, HDL, and lipoprotein-deficient plasma (LPDP) eluted.

**Figure 3 biomolecules-10-00829-f003:**
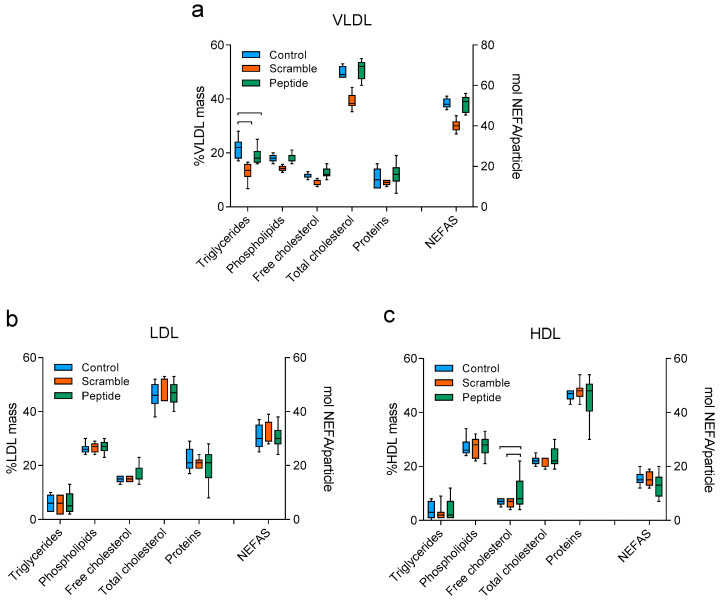
Lipoprotein composition. (**a**) VLDL, (**b**) LDL, (**c**) HDL. Each graph shows the relative mass of each major component (triglycerides, phospholipids, free cholesterol, total cholesterol, and proteins) and the amount in moles per lipoprotein particle of the non-esterified fatty acids (NEFAS). Data are shown as box-plot graphs, *n* = 8 (controls and scramble), *n* = 12 (peptide). Bars indicate *p* < 0.05 vs. other groups.

**Figure 4 biomolecules-10-00829-f004:**
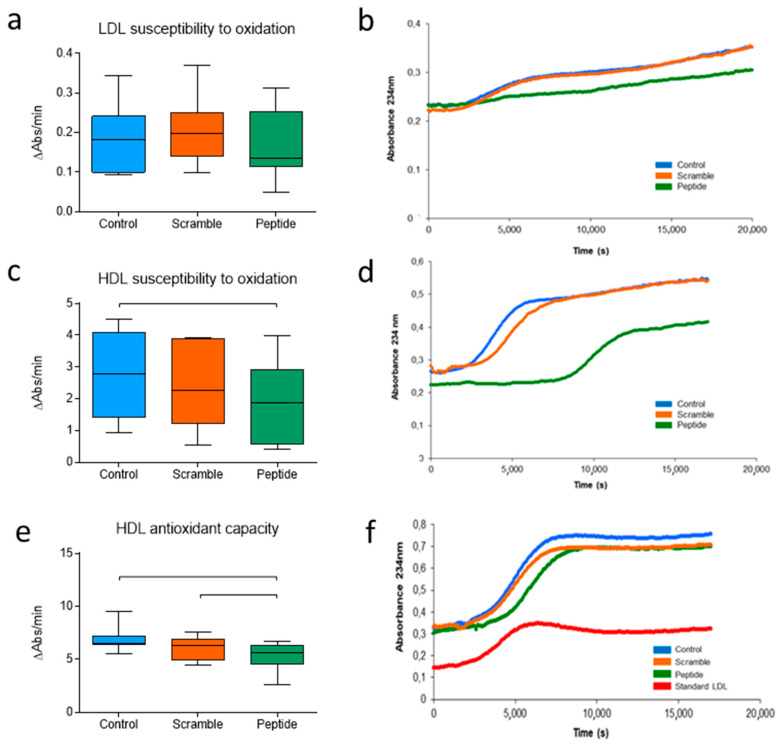
Oxidation-related parameters. Oxidation of lipoproteins dialyzed in PBS was induced by adding 5 µM CuSO_4_ and was monitored at 234 nm. (**a**) LDL susceptibility to oxidation. Data are expressed as the increase in the absorbance per minute. (**b**) Representative kinetics of LDL oxidation. (**c**) HDL susceptibility to oxidation. Data are expressed as the increase in absorbance per minute during the phase of maximal slope, mean ± SEM. (**d**) Representative kinetics of HDL oxidation. (**e**) Protective effect of HDL on LDL oxidation. Data are expressed as the increase in absorbance per minute during the maximal slope phase. (**f**) Representative kinetics of human LDL + murine HDL oxidation. Data in (**a**,**c**,**e**) are shown as box-plot graphs, *n* = 7 in control and scramble groups, *n* = 10 in the peptide group. Bars indicate *p* < 0.05 vs. other groups.

**Figure 5 biomolecules-10-00829-f005:**
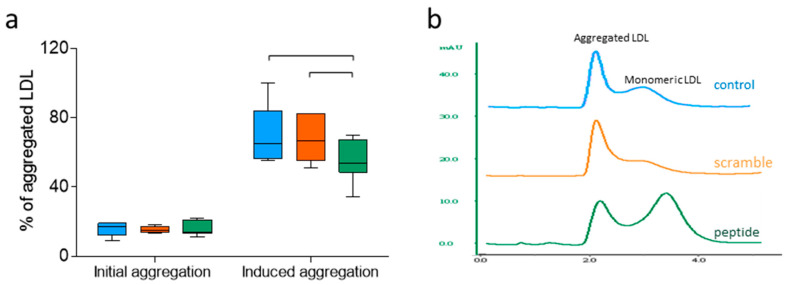
LDL susceptibility to aggregation. Susceptibility to aggregation experiments were conducted as indicated in the Methods. The degree of aggregation was estimated by size exclusion chromatography in a Superose 6 column, and the percentage of aggregated LDL particles was determined by quantifying the area of the eluted peaks corresponding to aggregated and non-aggregated LDL subfractions. (**a**) Percentage of aggregated LDL. Data are shown as box-plot graphs, *n* = 7 in control and scramble groups, *n* = 10 in the peptide group. Bars indicate *p* < 0.05 vs. other groups. (**b**) Representative chromatograms of LDLs from each group after aggregation.

**Figure 6 biomolecules-10-00829-f006:**
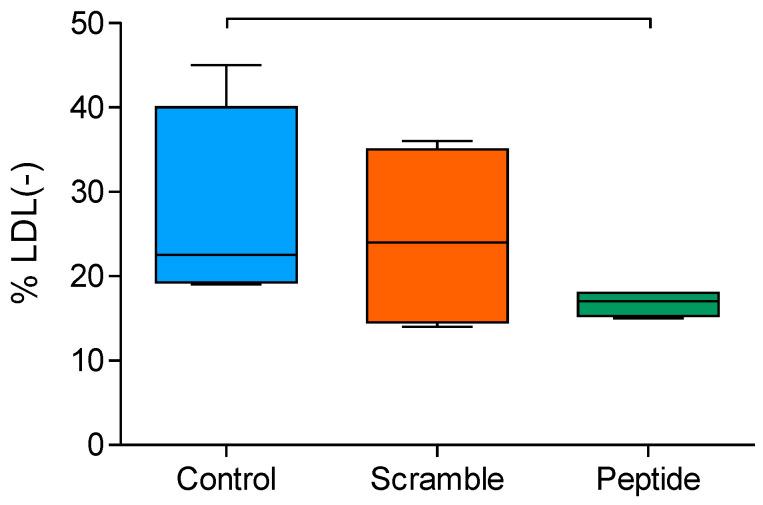
Electric charge of LDL. LDL dialyzed in buffer A was chromatographed in a MonoQ anion-exchange column, as described in the Methods. Data are expressed as the proportion of the electronegative LDL fraction. Data are shown as box-plot graphs, *n* = 4 in each group. Bars indicate *p* < 0.05 vs. other groups.

**Figure 7 biomolecules-10-00829-f007:**
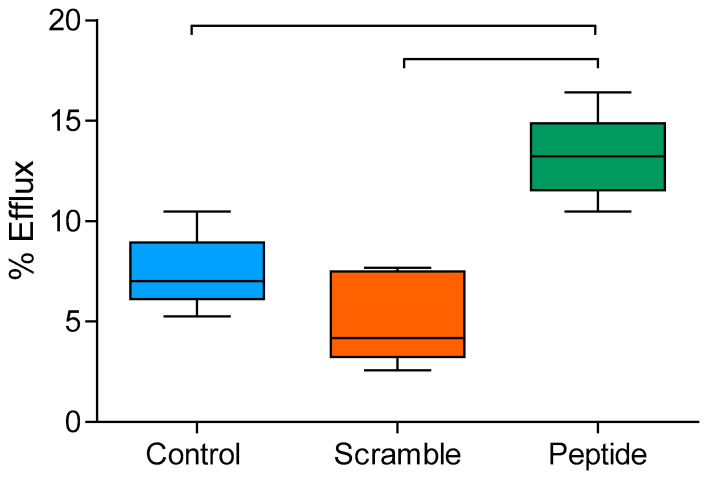
In vitro cholesterol efflux capacity of HDL. HDL (25 µg/mL) was incubated for 4 h with 3H-cholesterol-loaded J774A.1 mouse macrophages, as described in the Methods. Cholesterol efflux is expressed as the proportion of [^3^H]-cholesterol detected in the culture medium relative to total [^3^H]-cholesterol in cells and medium. Data are expressed as shown in box-plot graphs, *n* = 4 in the control and scramble group, with 6 mice in the peptide group. Bars indicate *p* < 0.05 vs. other groups.

**Figure 8 biomolecules-10-00829-f008:**
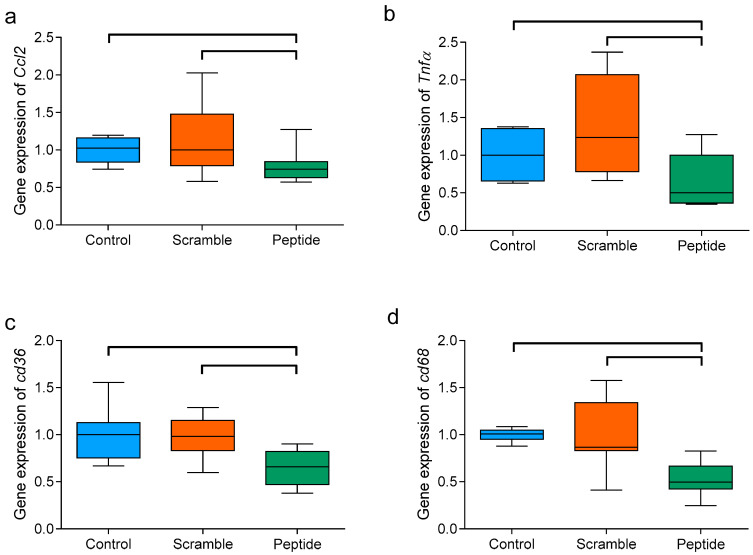
Hepatic expression of inflammation-related genes. (**a**) *Ccl2*, (**b**) *Tnfα*, (**c**) *Cd36*, and (**d**) *Cd68*. Total liver RNA was obtained as described in the Methods, and the expression of target genes was analyzed by RT-PCR. Data are expressed as shown in the box-plot graphs, *n* = 6 in each group. Bars indicate *p* < 0.05 vs. other groups.

**Figure 9 biomolecules-10-00829-f009:**
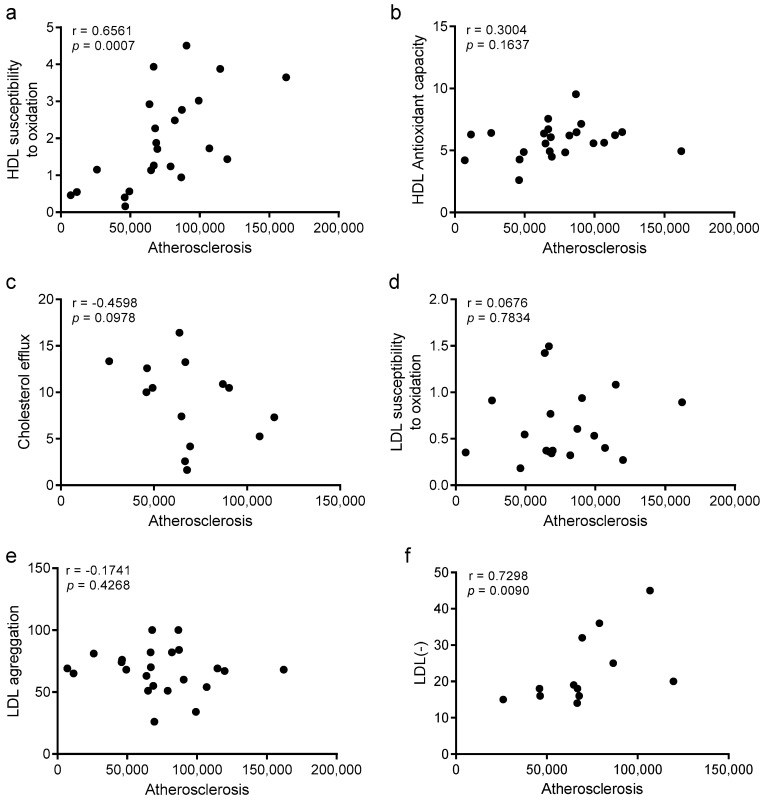
Univariate correlations between atherosclerosis development and lipoprotein function parameters. (**a**) HDL susceptibility to oxidation, (**b**) HDL antioxidant capacity, (**c**) cholesterol efflux induced by HDL, (**d**) LDL susceptibility to oxidation, (**e**) LDL susceptibility to aggregation, and (**f**) electronegative charge of LDL. Data were analyzed using Spearman’s rho correlation.

**Figure 10 biomolecules-10-00829-f010:**
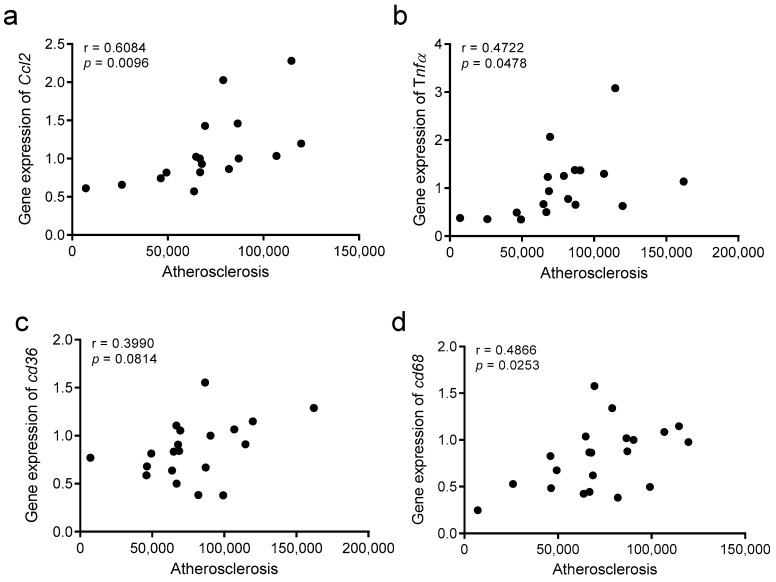
Univariate correlations between atherosclerosis development and hepatic inflammation parameters. (**a**) *Ccl2*, (**b**) *Tnfα*, (**c**) *Cd36*, (**d**) *Cd68*. Data were analyzed using Spearman’s rho correlation.

**Table 1 biomolecules-10-00829-t001:** Lipid profile and hepatic parameters of low-density lipoprotein receptor knockout mice (LDLR-KO) after eight weeks on a western diet.

	Controls	Scramble	d-[113–122]apoJ
Total cholesterol (mM)	52.5 ± 8.2	50.7 ± 7.7	51.9 ± 7.6
VLDL-c (mM)	39.3 ± 8.2	36.5 ± 7.5	34.9 ± 8.0
LDL-c (mM)	9.6 ± 2.0	10.4 ± 3.8	13.6 ± 4.0
HDL-c (mM)	3.5 ± 1.5	3.3 ± 0.9	3.5 ± 1.8
Triglycerides (mM)	9.6 ± 2.2	7.8 ± 3.1	7.9 ± 3.5
Phospholipids (mM)	10.8 ± 0.5	10.6 ± 1.5	10.2 ± 1.8
AST (U/L)	107.4 ± 71.4	110.3 ± 70.9	86.8 ± 47.6
ALT (U/L)	29.3 ± 20.2	38.4 ± 25.5	22.1 ±11.0
Liver weight (g)	1.48 ± 0.23	1.34 ± 0.32	1.33 ± 0.22
Liver cholesterol (µmol/g liver)	9.50 ± 4.67	11.57 ± 4.25	10.84 ± 3.63
Liver triglycerides (µmol/g liver)	12.42 ± 3.98	15.43 ± 3.05	11.51 ± 3.89
Liver phospholipids (µmol/g liver)	4.62 ±1.97	4.30 ± 1.75	4.64 ±1.43

Data are expressed as mean ± SD, *n* = 8 (controls and scramble), *n* = 12 (peptide). VLDL: very-low-density lipoprotein, LDL: low-density lipoprotein, HDL: high-density lipoprotein, AST: aspartate transaminase, ALT: alanine transaminase, SD: standard deviation.

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
