# Peer review of "Subcutaneous Administration of Apolipoprotein J-Derived Mimetic Peptide d-[113–122]apoJ Improves LDL and HDL Function and Prevents Atherosclerosis in LDLR-KO Mice"

_biomolecules, 2020, doi:10.3390/biom10060829_

Round 1

Reviewer 1 Report

The authors indicated subcutaneous administration of apoJ mimetic peptide, D-[113-122]apoJ, improved anti-oxidation of LDL and reduced atherosclerosis lesion of western diet-fed LDL receptor null mice. The 6th helix of apoJ was a substitute to D amino acid which retains the structure character but biological degradation is slow comparing peptides with L-amino acids. In this project, the western diet-fed LDL receptor null mice were used. By administration of apoJ mimetic peptide protect atherosclerosis legion in mice. 

  • Line 23, and line 85 G* amphipathic helix  is Class G* amphipathic helix
  • Line 272 more capacity—higher capacity
  • Line 260-1, Line 302-3, Line 324, Figure 6, Figure 7, and Line 346-7 https://www.graphpad.com/support/faq/what-is-the-meaning-of--or--or--in-reports-of-statistical-significance-from-prism-or-instat/

It is recommended to use asterisks to show the significance. If you wish to use the bar as it is, please describe, “Bar indicates statistically significant difference, P<005.” in the legends.

  • Line 114, Navab and al reposted,  is this "et al"?
  • Line 226, No statistical difference? Please describe.
  • Line 230, almost identical, what is different? Please be specific.

Minor points:

  • Fig2, fig 4, and Fig5 color of control are different light blue and dark blue. Please use an identical color to indicate one experimental group.
  • Line 268 Fig 4a, 4b Line 271 Fig 4c, 4d Line 273 Fig 4e, 4f

Line 382 Please match the character of C) and D) as in the figure.  , c) and d)

  • Fig4 f introductory remark is located over the data line(red), you need to change the location

Author Response

REVIEWER 1

The authors indicated subcutaneous administration of apoJ mimetic peptide, D-[113-122]apoJ, improved anti-oxidation of LDL and reduced atherosclerosis lesion of western diet-fed LDL receptor null mice. The 6th helix of apoJ was a substitute to D amino acid which retains the structure character but biological degradation is slow comparing peptides with L-amino acids. In this project, the western diet-fed LDL receptor null mice were used. By administration of apoJ mimetic peptide protect atherosclerosis lesion in mice.

We thank the reviewer for his/her kind appreciation of our work and for helpful comments. The point-by-point response to these comments is below.

Line 23, and line 85 G* amphipathic helix  is Class G* amphipathic helix

Line 272 more capacity—higher capacity

 Changes have been included according to the reviewer’ suggestions

Line 260-1, Line 302-3, Line 324, Figure 6, Figure 7, and Line 346-7 https://www.graphpad.com/support/faq/what-is-the-meaning-of--or--or--in-reports-of-statistical-significance-from-prism-or-instat/

It is recommended to use asterisks to show the significance. If you wish to use the bar as it is, please describe, “Bar indicates statistically significant difference, P<005.” in the legends.

We agree with the reviewer that asterisks or other symbols are more frequently used to indicate statistical differences. However, we consider that, in our case, perhaps is clearer to indicate differences with bars. In all legends is included “Bars indicate P<0.05 vs other groups”

Line 114, Navab and al reposted,  is this "et al"?

Changes have been included according to the reviewer’ suggestion.

Line 226, No statistical difference? Please describe.

“no statistical difference in the parameters of the lipid profile was observed between groups” has been included according to the reviewer’ suggestion (line 229).

Line 230, almost identical, what is different? Please be specific.

We mean that the lipid profile chromatograms overlapped, either by measuring the absorbance at 280 nm or the cholesterol content. Statistical analysis was not performed in these assays because only 3 plasmas of each group were analyzed, but all the chromatograms were very similar. The sentence has been modified according to the reviewer’ suggestion (line 233).

Minor points:

Fig2, fig 4, and Fig5 color of control are different light blue and dark blue. Please use an identical color to indicate one experimental group.

The color dark blue in the chromatograms of these figures has been changed to light blue.

Line 268 Fig 4a, 4b Line 271 Fig 4c, 4d Line 273 Fig 4e, 4f

Line 382 Please match the character of C) and D) as in the figure.  , c) and d)

Thanks for your kind remark, we have detected the same mistake in other parts of the document. Now, the text has been modified according to the reviewer’ suggestion

Fig4 f introductory remark is located over the data line(red), you need to change the location

Thanks for the comment, the figure has been modified according to the reviewer.

Reviewer 2 Report

The paper is potentially interesting since it focuses on the use of one peptide acting simultaneously on the function of both HDL and LDL in regard the onset or reducing the development of atheromatous lesions. The topic is interesting and scientifically important, but there are several points that need to be explained and revised before it is acceptable.

  • Sexual dimorphism in metabolic disorders such as obesity and atherosclerosis are well described and biologically important element of understanding their aetiologies. Historically, females were thought to have greater variability due to the oestrogen cycle. Why did the authors use female mice to perform their experiments? What occur in male mice?
  • All mice received atherogenic diet. It’s could be better to use also mice fed with chow diet as control.
  • Please, show data about weight gain and food intake of the different groups during the 8-week diet.
  • Figure 1b and 1c do not fully reflect the quantification in figure 1a. Could the authors show more representative images?
  • In figure 4, 5 and 6 there are no statistically differences between scramble and peptide, please explain the reasons.
  • Authors should briefly explain why they have chosen Ccl2 and Tnfalpha as markers of hepatic inflammation. What about some interleukins involved? What about F4/80 expression in liver? The use of immunohistochemistry is more suitable rather the evaluation of gene expression.

Author Response

REVIEWER 2

The paper is potentially interesting since it focuses on the use of one peptide acting simultaneously on the function of both HDL and LDL in regard the onset or reducing the development of atheromatous lesions. The topic is interesting and scientifically important, but there are several points that need to be explained and revised before it is acceptable.

We thank the reviewer for his/her appreciation of our work and for helpful critique. The point-by-point response is below.

Sexual dimorphism in metabolic disorders such as obesity and atherosclerosis are well described and biologically important element of understanding their aetiologies. Historically, females were thought to have greater variability due to the oestrogen cycle. Why did the authors use female mice to perform their experiments? What occur in male mice?

The main reason for using females is that the development of atherosclerosis is higher in mice females than in males (Mansukhani NA, Wang Z, Shively VP, Kelly ME, Vercammen JM, Kibbe MR. Artery Res. 2017;20:8–11. doi:10.1016/j.artres.2017.08.002), as is stated in the current version of the manuscript (lines 102-103). Although we think that if the study would have been conducted in males we would have obtained the same results, we considered that the best was to use the gender in which the development of lesions is higher.

All mice received atherogenic diet. It’s could be better to use also mice fed with chow diet as control.

In the same line of the previous question, only chow diet “per se” does not promote atherosclerosis in LDLR-KO mice; therefore, the inhibitory effect of the peptide could have not been detected.

Please, show data about weight gain and food intake of the different groups during the 8-week diet.

These data have been included in the supplementary data as Figure S1. Of note, we observed that the administration of D-[113-122]apoJ promoted a delay in the weight gain. The reason of this effect is unknown, but it was not due to differences in food intake. We consider that this finding is very intriguing and merits further research. In accordance, we want to conduct a new study focused on the effect of the peptide on the accumulation of fat and the involved molecular mechanisms. However, the study has not started due to the coronavirus crisis and we will have to wait for several months before obtaining some results. Anyway, we think that this study on fat accumulation is beyond the scope of the current manuscript, which is focused on atherosclerosis development.

Figure 1b and 1c do not fully reflect the quantification in figure 1a. Could the authors show more representative images?

Thanks for the comment. Perhaps the problem was that, in order to preserve the square form of the figure, we included only a part of the aorta with the bulk of the lesions. But it is more correct to include the whole aorta, as is shown in the modified version of the figure. The ratio of the areas of lesions of the selected images in 1b (control 68472 µm2/section), 1c (scramble 67495 µm2/section) and 1d (peptide 35965 µm2/section) (ratio control/peptide 1.90) is similar to the ratio of areas shown in the graph 1a (control 83283,9 µm2/section, scramble 85150,4 µm2/section and peptide 47205,70 µm2/section) (ratio control/peptide 1.76).

In figure 4, 5 and 6 there are no statistically differences between scramble and peptide, please explain the reasons.

The reason for the lack of differences between peptide and scramble is unclear. In some parameters the scramble seems to have an intermediate behavior between control and peptide, although it is always much closer to the control mice than to the peptide-treated mice. Probably, increasing the number of mice in each group, statistical differences between peptide and scramble would be achieved. However, a marginal effect of the scramble cannot be definitely discarded.

Authors should briefly explain why they have chosen Ccl2 and Tnfalpha as markers of hepatic inflammation. What about some interleukins involved? What about F4/80 expression in liver? The use of immunohistochemistry is more suitable rather the evaluation of gene expression.

Ccl2 and TNFa are representative of hepatic inflammation. Of course, other cytokines such as IL6, IL8 or IL1b or molecules as CRP could had been used. However, in our experience most of the inflammatory cytokines in the liver respond in a similar way to pro-inflammatory/anti-inflammatory stimuli, and we considered that Ccl2 and TNFa are representative of the inflammatory response. Hepatic elevations in Ccl2, which is involved in macrophage infiltration, and TNFa, which is produced by activated macrophages,  are frequently used to detect  hepatic inflammation (Miura K, Yang L, van Rooijen N, Ohnishi H, Seki E. Hepatic recruitment of macrophages promotes nonalcoholic steatohepatitis through CCR2. Am J Physiol Gastrointest Liver Physiol. 2012;302(11):G1310–G1321). The sentence “Because mRNA levels of Tnfα, Ccl2, and Cd68  are commonly elevated in hepatic inflammation, they were selected to assess the potential anti-inflammatory response by ApoJ mimetic peptide in this organ”  has been included in the Discussion section (page 14, lines 451-453)).

Regarding F4/80 (EMR1 in humans), we appreciate the suggestion of the reviewer, but we have never measured the expression of this gene, which would be a marker similar to CD68, indicating the infiltration of macrophages. Interestingly, it has been reported that F4/80 and CD68 liver expression decreases after the administration of apoA-I-derived peptides (PLoS One. 2020 Jan 8;15(1):e0226931); the decrease of the expression of both molecules was very similar. However, it is a worthy suggestion to be considered for future studies.

Regarding immunohistochemistry, unfortunately we have no fixed hepatic tissue to conduct these experiments.

Round 2

Reviewer 2 Report

Thank you for your revision of this manuscript. I appreciate the time that you have put into revising your manuscript based upon my comments.

Author Response

Thank you for your kind comments